# Full-Length Transcriptome Sequencing Analysis and Characterization of WRKY Transcription Factors Responsive to Cadmium Stress in *Arabis paniculata*

**DOI:** 10.3390/plants12213779

**Published:** 2023-11-06

**Authors:** Tianjiao Chen, Dan Zuo, Jie Yu, Yunyan Hou, Hongcheng Wang, Lei Gu, Bin Zhu, Huinan Wang, Xuye Du

**Affiliations:** School of Life Sciences, Guizhou Normal University, Guiyang 550025, China; 222100100410@gznu.edu.cn (T.C.); 21010100449@gznu.edu.cn (D.Z.); 21010100402@gznu.edu.cn (J.Y.); 21010100376@gznu.edu.cn (Y.H.); wanghc@gznu.edu.cn (H.W.); leigu1216@nwafu.edu.cn (L.G.); zhugg130@126.com (B.Z.)

**Keywords:** *Arabis paniculata*, full-length transcriptome, WRKY family, molecular characterization, expression analysis, Cd responsive

## Abstract

*Arabis paniculata* is a newly discovered hyperaccumulator known for its ability to accumulate multiple metals. WRKY proteins play a significant role in plant responses to various stresses, including cadmium (Cd) stress. However, there is limited research on the molecular biology of *Arabis paniculata*, especially regarding the WRKY family. In this study, we conducted third-generation sequencing for functional annotation and structural analysis of *Arabis paniculata.* We obtained 41,196 high-quality isoforms from the full-length transcriptome, with an average length of 1043 bp. A total of 26,670 genes were predicted against NR, Swissprot, KOG, and KEGG databases. Functional comparison using the KOG database revealed excellent annotation in 25 functional categories, with general function prediction (1822 items) being the most predominant. MISA analysis identified 12,593 SSR loci, with single nucleotide repeats being the largest category (44.83% of the total). Moreover, our predictions provide insights into 20,022 coding sequences (CDS), 811 transcription factors, and 17,963 LncRNAs. In total, 34 *WRKY* gene sequences were identified in *Arabis paniculata.* Bioinformatics analysis revealed diverse numbers of amino acids in these WRKYs (113 to 545 aa), and a conserved WRKYGQK sequence within the N-terminus of the WRKY protein. Furthermore, all WRKYs were found to be localized in the nucleus. Phylogenetic analysis classified the *WRKY* genes into three categories: I (14 members), II (17 members), and III (3 members). Category II was subsequently divided into four sub-categories: II-a (8 members), II-b (1 member), II-c (1 member), and II-d (7 members). Our quantitative real-time polymerase chain reaction (qRT-PCR) experiments revealed that *ApWRKY23* and *ApWRKY34* exhibited the highest expression levels at the 24-h time point, suggesting their potential role as the candidate genes for Cd stress response. These findings contribute to our understanding of the genomic information of *Arabis paniculata* and provide a basis for the analysis of its genetic diversity. Additionally, this study paves the way for a comprehensive exploration of the molecular mechanisms underlying the *WRKY* genes in *Arabis paniculata* under Cd stress conditions.

## 1. Introduction

Cd is a non-essential and toxic heavy metal that poses serious risks. Its toxicity impedes the ability of plants to absorb water and essential nutrients, which in turn has negative effects on photosynthesis, transpiration rate, and chlorophyll pigmentation. Consequently, these adverse effects lead to a range of detrimental changes in plant morphology, growth, development, physiology, biochemistry, and structure. Some of these changes include growth retardation, water imbalance, and delayed seed germination, ultimately resulting in significant agricultural losses [1,2,3,4]. As one of the most toxic heavy metal pollutants, Cd has consistently emerged as the primary contaminant in agricultural soils worldwide. The severity of Cd pollution continues to escalate [5]. Cd toxicity has received growing attention due to its potential for plant contamination, leading to disruption of cell redox control and interference with the production of reactive oxygen species (ROS) [6]. However, under conditions of heavy metal stress, the production of ROS may exceed the scavenging capacity of antioxidants, resulting in oxidative stress and destruction of biomolecules [7]. These substances can cause damage to cells through various mechanisms, such as DNA reactions triggering gene mutations, alteration of protein side chain structures, and depletion of phospholipid molecules in cell membranes [8].

*Arabis paniculata* has the ability to accumulate heavy metal ions, including Cd, lead (Pb), and zinc (Zn) [9]. In a previous study, our group made significant findings regarding the molecular and physiological processes involved in heavy metal stress tolerance in *Arabis paniculata*, which facilitates the further utilization of *Arabis paniculata* in phytoremediation [10]. However, the current understanding of the molecular biology of *Arabis paniculata* is still limited, which greatly hampers in-depth investigations into its growth and development, metabolic regulation, resource utilization, and other aspects.

Gene transcriptional regulation is widely recognized as the foundation of plant germination, growth, and tissue differentiation, with numerous transcription factors (TFs) playing indispensable roles in these processes [11]. Among these TFs, WRKY proteins contribute significantly to various physiological mechanisms within plant cells. This protein family shares a highly conserved domain consisting of 60 amino acid residues, including the conserved WRKYGQK sequence, followed by a zinc finger motif located in the C-terminal region of the protein [12]. The WRKY represents a distinctive class of plant TFs characterized by a highly conserved WRKYGQK amino acid sequence [13]. Originally discovered in sweet potato [14], studies have demonstrated that WRKY proteins are engaged in diverse physiological processes, both under normal growth conditions and when subjected to various stressors [15].

The functionality of WRKY TFs relies on their ability to recognize and bind to specific W-box [(T)TGAC(C/T)] sequences, thereby regulating the expression of target genes precisely [16]. Based on the number of conserved domains and the structural characteristics of their zinc finger motifs, WRKY TFs can be classified into three categories: Class I proteins possess two conserved WRKY domains and feature a C_2_H_2_ zinc finger structure, while classes II and III contain a single conserved WRKY domain. Class II exhibits a C_2_H_2_ zinc finger structure, and class III features a C_2_HC structure. Moreover, class II can be further subdivided into five subclasses (IIa, IIb, IIc, IId, and IIe) based on variations in amino acid sequences outside the conserved structure [17].

Studies have unequivocally demonstrated the essential role of WRKYs in a range of physiological processes, encompassing embryonic development [18], root growth [19], seed germination and dormancy [20], senescence [21], as well as participation in abiotic stress, such as heavy metal exposure [22]. However, the identification and bioinformatics analysis of the *WRKY* gene family in *Arabis paniculata* has not yet been reported. In this study, we utilized the PacBio molecular real-time sequencing platform, a third-generation sequencing technology, for column splicing assembly, functional annotation, and classification. Building upon the full-length transcriptome data of *Arabis paniculata*, this study employed bioinformatics methods to identify and analyze its WRKY TFs. Additionally, we examined the relative expression of *WRKY* gene family members in *Arabis paniculata* under Cd stress using the quantitative real-time polymerase chain reaction (qRT-PCR) method. These findings lay the groundwork for future studies focusing on the molecular mechanisms underlying the response of *Arabis paniculata*’s *WRKY* gene family members to Cd stress.

## 2. Results

### 2.1. Transcriptome Sequencing of Arabis paniculata

The raw sequence data of full-length transcriptome are available in BioProject (https://ngdc.cncb.ac.cn/bioproject/, accessed on 9 October 2022) under the accession number PRJCA002559, accessed on 21 April 2020. The transcriptome analysis of *Arabis paniculata* seedlings was conducted to obtain comprehensive sequencing data. The results revealed a total of 39,652,440 subreads, with a cumulative base count of 34,400,084,025 bp. The average length of the subreads was 867 bp, and the N50 length was 1164 bp. After applying quality filters (full passes ≥ 1), 41,432 high-precision CCS sequences were obtained for subsequent transcript analysis. Following CCS clustering, correction, and redundancy removal steps, a set of 41,196 high-quality isoforms (QV > 99%) were obtained. These isoforms had an average length of 1043 bp, an N50 length of 1613 bp, and a GC content of 44%. The exceptional quality of the sequencing data provides a reliable standard of reference for functional annotation and structural analysis of the plant species.

### 2.2. Four Database Function Annotation

The 41,196 isoforms were analyzed using BLASTX and four databases (NR, KEGG, KOG, and SwissProt) were used to compare consistency. The results showed that a total of 26,670 genes (64.74%) were annotated (Figure 1A). Among these databases, the NR database had the highest success rate of annotation, with 24,794 genes (92.97%) being annotated. The SwissProt database followed with 20,847 (78.17%), while the KEGG database was next with 14,174 (53.15%) successful annotations. The KOG database had the lowest rate of success, with only 11,081 (41.55%) genes being annotated successfully. In total, 7803 genes, accounting for 29.26%, were annotated using all four databases.

### 2.3. GO Function Annotation Analysis

To obtain the GO information for each transcript, the GO annotation was performed on the full-length transcript, followed by secondary classification statistics. GO function annotation is classified into three categories: Biological Process, Cellular Component, and Molecular Function (Figure 1B). Under the Biological Process category, there are a total of 22 categories. The category with the highest number of entries is Cellular Process, which accounts for 16,051 entries. Metabolic Process follows closely with 15,078 entries, while the category with the fewest entries is represented by only 5 transcripts. In the Cellular Component category, there are 19 categories in total. The category with the highest number of entries is Cellular, with 21,308 transcripts assigned to this category. Cell Part is the second highest, with 21,302 entries. On the other hand, Other Organism and Other Organism Part have the fewest entries, each represented by only 1 transcript. Within the Molecular Function category, there are 11 categories. Binding stands out as the most prevalent category with 14,112 entries. Catalytic Activity comes next with 11,087 entries, while Metallochaperone Activity represents the least common category, comprising only 5 transcripts.

### 2.4. KOG Function Annotations

The KOG database annotated a total of 11,081 genes into 25 functional groups (Figure 1C). Among these groups, the one with the highest number of annotated genes was General Function Prediction Only, which includes 1822 genes from *Arabis paniculata*. Circular genes were annotated to Posttranslational Modification, Protein Turnover, and Chaperones, totaling 1603. Signal Transduction Mechanisms had annotations for 973 genes. Extracellular Structures had at least 21 annotations, while Cell Motility did not receive any annotations.

### 2.5. KEGG Metabolic Pathway Annotation Analysis

The KEGG database was utilized for pathway annotation. The metabolic pathway can be classified into five branches: Cellular Processes (A), Environmental Information Processing (B), Genetic Information Processing (C), Metabolism (D), and Organismal Systems (E). A total of 14,174 unigenes were successfully mapped to the KEGG database (Figure 1D). Within the pathways, 759 genes were involved in the Cellular Processes pathway, 389 genes were associated with the Environmental Information Processing pathway, and 2443 genes were related to the Genetic Information Processing pathway. The Metabolism pathway encompassed a total of 8847 genes distributed across 11 categories. As for Organismal Systems, it involved 434 genes within a single category. Moreover, Translation and Carbohydrate Metabolism exhibited the highest gene count, with 1856 and 1717 genes, respectively.

### 2.6. Advanced Annotation Analysis

The analysis yielded 20,022 CDS sequences (Figure 2), with gene lengths ranging from 99 to 2361 bp. Plant TFs were predicted using the iTAK database, resulting in a total of 56 TFs families. The largest family included *AP2/ERF-ERF* with 97 members, followed by *GRAS* with 56 members and *bHLH* with 52 members.

### 2.7. SSR Analysis

MISA (version 1.0, default parameters) was employed to detect SSR loci in the full-length transcriptome of *Arabis paniculata*. A total of 12,539 SSR loci were identified (Figure 3 and Appendix A), including 5621 single nucleotide repeats, 3642 dinucleotide repeats, 3116 trinucleotide repeats, 71 tetranucleotide repeats, 23 pentanucleotide repeats, and 66 hexanucleotide repeats, accounting for 44.83%, 29.05%, 24.85%, 0.57%, 0.18%, and 0.53% of the total SSRs, respectively. Among them, the A/T repeat motif was the dominant motif in single nucleotide repeats, accounting for 98.22% of the total number of single nucleotide repeats. The AG/CT motif was the dominant repeat motif in dinucleotide repeats, accounting for 47.96% of the total dinucleotide repeats. AAG/CTT was the dominant repeat motif in trinucleotide repeats, constituting 46.89% of the total trinucleotide repeats. AAAG/CTTT was the dominant motif in tetranucleotide repeats, accounting for 29.58% of the total number of tetranucleotide repeats. These results provide a foundation for the development of molecular marker technology in *Arabis paniculata*.

### 2.8. LncRNA Analysis

LncRNAs in *Arabis paniculata* were predicted using the CPAT software, which efficiently distinguished coding and non-coding transcripts among numerous candidates. The analysis identified a total of 17,963 LncRNAs (Figure 4).

### 2.9. Identification of WRKY Gene in Arabis paniculata

A total of 36 *WRKY* sequences were identified in *Arabis paniculata* based on the full-length transcriptome data. After database alignment and removal of redundant, repetitive, and incompletely annotated TFs, 34 *WRKY* family members were ultimately identified and named *ApWRKY1*–*ApWRKY34*.

### 2.10. Physicochemical Properties Analysis of WRKY Protein Sequences in Arabis paniculata 

The physicochemical properties of the 34 identified *WRKY* gene family members in *Arabis paniculata* were analyzed, including amino acid number, protein molecular weight (MW), theoretical isoelectric point (*p*I), instability coefficient, and average hydrophobicity index (Appendix A). The findings revealed that the amino acids ranged from 113 to 545, with the relative MW of the proteins varied from 12,650.49 to 60,298.21 Da, with an average of 36,965.46 ku. The *p*I ranged from 5.49 to 9.98, with an average of 8.14. Hydrophilicity analysis indicated that all the proteins exhibited negative hydrophilicity values, indicating their classification as hydrophilic proteins. ApWRKY14 exhibited the highest hydrophilicity with an average value of −0.834, while ApWRKY18 displayed the highest hydrophobicity with a value of −0.492. The instability index values of ApWRKY34, ApWRKY32, ApWRKY22, and ApWRKY21 were less than 40, indicating protein stability. The remaining 30 members of the gene family were classified as unstable proteins with a short lifespan. Based on the result of subcellular prediction, all 34 WRKY genes were found to be located in the nucleus.

### 2.11. Protein Structure Analysis of WRKYs in Arabis paniculata 

The protein structure analysis of the *WRKY* gene family in *Arabis paniculata* was conducted (Appendix A). The ApWRKYs exhibited four main secondary structures: α-helix, β-sheet, random coil, and extended chain. The random coil structure was the most prevalent, comprising 50.2% to 79.76% of the total. The α-helix followed, ranging from 7.34% to 33.64%. The β-sheet accounted for 6.54% to 18.78%, while the extended chain was relatively minor, ranging from 0.88% to 6.76%. The results of the protein structure analysis demonstrated a high level of similarity in *Arabis paniculata*. Due to different folding methods, these proteins may exhibit variation, which can result in a wide range of biological functions.

### 2.12. Multiple Sequence Alignment and Phylogenetic Analysis

The multiple alignments of ApWRKY protein sequences revealed that the N-terminal WRKY domain of all 34 WRKY proteins contained a conserved sequence of WRKYGQK. Additionally, the C-terminal region exhibited a typical zinc finger structure, either C_2_H_2_ or C_2_HC (Figure 5A). Based on the classification principle of the WRKY family, the ApWRKYs were categorized into three classes (Figure 5B). Category I included 14 WRKY proteins with two WRKYGQK domains and one C_2_H_2_ (CX_4_CX_23_HXH) zinc finger structure. Class II consisted of 17 WRKY proteins with a single WRKYGQK domain and a single C_2_H_2_ (CX_4_CX_23_HXH) zinc finger structure. Class III comprised three WRKY proteins with only one WRKYGQK domain and one C_2_HC (CX_4_CX_23_HXC) zinc finger structure.

### 2.13. Identification of Conserved Motifs in WRKY Genes of Arabis paniculata

After analyzing the conserved domain of the WRKY protein in *Arabis paniculata*, 10 motifs were selected using the online tool MEME for visual examination (Figure 6A). It is evident that different WRKY proteins exhibit varying numbers of motifs. Notably, Motif 1 and Motif 3 contain a complete WRKY conserved domain, with Motif 1 being present in all sequences except ApWRKY33. This suggests that Motif 1 has been highly conserved throughout evolution. Composition and distribution patterns of WRKY within the same subgroup were similar, although variations in motif number and distribution were observed across different subgroups. Furthermore, a higher amino acid residue character was indicative of a larger ordinate in each motif diagram, implying greater frequency and higher conservation and stability of the motifs (Figure 6B). 

### 2.14. Conserved Domain Analysis of WRKY Genes in Arabis paniculata

To analyze the conserved domains within the identified WRKYs of *Arabis paniculata*, the online domain analysis software provided by NCBI was utilized. The analysis revealed that all sequences contained the core motif of the WRKY domain, with some WRKY proteins harboring two WRKY domains, primarily found in the first group. Notably, seven members of the gene family contained the Plant_zn_clust domain (Figure 7). These structural variations may contribute to the functional divergence observed among WRKY proteins.

### 2.15. Expression Profiles of ApWRKYs in Response to Cd Stress

To investigate the expression patterns of WRKY genes in the above-ground parts of *Arabis paniculata* under Cd stress, qRT-PCR was performed at various time points (0, 0.5, 6, and 24 h) using *Arabis paniculata* subjected to 5 mM Cd condition (Figure 8). As the duration of Cd stress increased, the expression of *WRKY* genes exhibited diverse trends. Specifically, the expression levels of *ApWRKY12*, *ApWRKY31*, and *ApWRKY32* decreased steadily, while *ApWRKY23*, and *ApWRKY34* displayed continuous increases. Notably, after 0.5 h of Cd stress, *ApWRKY19* demonstrated the highest expression level, approximately eight times that of the control (0 h). After 24 h of Cd stress, both *ApWRKY23* and *ApWRKY34* exhibited the highest expression levels, approximately 27 and 50 times higher than the control, respectively. These findings indicate that *ApWRKY23* and *ApWRKY34* represent promising candidate genes associated with tolerance to Cd stress.

## 3. Discussion

At present, for the sequencing of full-length transcriptome, PacBio single molecule real-time sequencing technology (SMRT) with long read and long sequencing characteristics is mainly used [23]. SMRT has been successfully used in the study of a variety of plants [24,25]. SMRT sequencing technology provides a complete sequence of new cognition, which has been proved to play a positive role in gene annotation and interpretation of gene function, especially for species without reference genome [26,27]. Transcriptome sequencing is an efficient and effective technology that can generate a large number of sequence data. These large number of cDNA sequences provide valuable information resources for genomics and genetics research [28,29,30,31,32]. In this study, we sequenced and analyzed the full-length transcriptome of *Arabis paniculata* using third-generation sequencing technology. This resulted in 41,196 high-quality transcripts with an average length of 1043 bp. Comparative analyses were conducted using the NR, KEGG, KOG, and Swissprot databases, revealing that the annotations of the *Arabis paniculata* transcriptome accounted for 92.97%, 53.15%, 41.55%, and 78.17% of the annotated transcripts, respectively. The classification analysis of GO functional annotations demonstrated that the properties of the *Arabis paniculata* transcriptome were primarily associated with cellular and metabolic processes. Additionally, the analysis of KEGG pathway annotations further supported the dominance of metabolic processes. Furthermore, through SSR analysis of the *Arabis paniculata* transcriptome data, we obtained a total of 12,593 SSR sites. Among these, single nucleotide repeat sites were the most abundant, followed by dinucleotide repeat sites. Moreover, the annotation of the *Arabis paniculata* transcriptome led to the identification of 811 TFs belonging to 56 TFs families. The WRKY TFs family members participate in a wide range of developmental and physiological processes, particularly in plant responses to various biotic and abiotic stresses [31,32,33,34]. This diverse family has been identified in multiple plant species, including *Arabidopsis thaliana*, swamp rice, tomato, and soybean, with respective counts of 72, 102, 81, and 197 *WRKY* genes [12,35,36,37,38]. However, no studies have investigated the *WRKY* gene family of *Arabis paniculata*. *Arabis paniculata*, in comparison to other species, exhibits a relatively smaller number of TFs. This difference suggests that the evolution of the *WRKY* gene family in *Arabis paniculata* may have been influenced by selective pressures exerted by external environmental factors. In conclusion, the WRKY TFs are key regulators in plant defense and signal transduction. They have been identified in various plant species, but their presence in *Arabis paniculata* remains unexplored. The lower count of TFs in *Arabis paniculata* hints at potential environmental influences on the evolution of the *WRKY* gene family in this particular species. Further research is warranted to unravel the specificities of the *WRKY* gene family in *Arabis paniculata* and its functional significance in plant responses to stress.

Combining phylogenetic trees from various TFs families offers valuable insights into both the evolutionary relationships among members and potential functional hypotheses based on identified branches [39]. The presence of the *WRKY* gene in prokaryotes suggests an ancient origin of the *WRKY* gene family that predates the diversification of plant phyla [40]. Based on the number of WRKY protein motifs and the variation in the zinc finger C_2_HH (C) domain, the *WRKY* gene family can be categorized into three groups: class I, class II, and class III. Notably, class II can be further subdivided into five subfamilies, namely classes IIa to IIe, as determined by phylogenetic analysis [12]. In our study, a total of 34 *WRKY* gene family members were identified. Among these, 14 members contained two WRKYGQK domains and exhibited a C_2_H_2_-type zinc finger structure (CX_4_CX_23_HXH), classifying them as class I. Additionally, 17 ApWRKYs harbored a single WRKYGQK domain and a C_2_H_2_-type zinc finger structure (CX_4_CX_23_HXH), and were classified under class II. Furthermore, the class II *ApWRKY* genes were further divided into five subclasses: II-a, II-b, II-c, II-d, and II-e, consisting of 8, 1, 1, 7, and 0 members, respectively. In terms of membership numbers, class II has the highest number of members, whereas class III has the lowest number of members. This distribution of membership aligns with the findings in the *Andrographis paniculata* [41]. Notably, three of these subclasses contained only a WRKYGQK domain and exhibited a C_2_HC-type zinc finger structure (CX_4_CX_23_HXC), thereby falling under class III. The loss of domain structure is regarded as a natural phenomenon and is also perceived as a driving force behind the divergence in gene family amplification [42,43,44]. A loss of zinc finger structure was observed in certain genes, possibly indicating a connection to gene evolution and sequencing quality. Further analysis of conserved sequences within the WRKY family in *Arabis paniculata* revealed the localization of the WRKY domain in Motif 1 and Motif 3. However, it was evident that Motif 1 and Motif 3 had distinct sequence structures, suggesting different functional characteristics.

WRKY TFs play a crucial role in the interconnected signaling pathways governing plant responses and development, particularly in relation to defense mechanisms against abiotic stress [16,45]. These TFs are instrumental in regulating plant tolerance to various abiotic stresses [46]. High levels of gene expression may indicate the pivotal role of plants under both abiotic and biotic stress conditions [47]. In this study, the response of *ApWRKY34* to Cd stress was investigated, and its expression level increased approximately 50-fold after 24 h of exposure, indicating its involvement in stress resistance mechanisms. Notably, the expression of *WRKY* genes often exert a significant influence on plant development [48]. The results of the expression analysis of *ApWRKYs* under different treatment durations further affirmvalidate the role of *WRKY* genes in *Arabis paniculata*’s response to Cd stress.

## 4. Materials and Methods

### 4.1. Plant Materials and Grow Conditions

The seeds of *Arabis paniculata* were preserved by our laboratory. In this experiment, *Arabis paniculata* seeds were sterilized with 2% sodium hypochlorite for 8 min and rinsed five times with ddH_2_O before being vernalized at 4 °C for three days. These seeds were cultivated in an artificial climate chamber under specific conditions to promote germination and growth. The chamber maintained a light/dark photoperiod of 14/10 h, a relative humidity of 60%, a light temperature of 26 °C, and a dark temperature of 21 °C for a duration of seven days. Seedlings were transplanted into a medium comprising sand and vermiculite in a ratio of 2:1, Hoagland nutrient solution was used for irrigation every day.

### 4.2. Construction of Full-Length Transcriptome cDNA Library

After 30 days of growth in the artificial climate chamber, the healthy plants of *Arabis paniculata* were randomly chosen, washed, and rapidly frozen in liquid nitrogen before being sent to a commercial corporation (Berry Genomics, Beijing, China) for full length transcriptome sequencing.

Total RNA was extracted from *Arabis paniculata* root and shoot samples using the EASY Rotating Plant RNA Extraction Kit (Aidlab, Beijing, China). RNA extracted from root and shoot tissues were mixed in equal proportions. The extracted total RNA underwent examination to assess degradation, contamination, purity, quantity, and integrity. Once the results met the required standards, the samples were thoroughly mixed and utilized for library construction and sequencing.

### 4.3. Identification of WRKY Family Members in Arabis paniculata 

The corresponding hidden Markov model (HMM) file for the conserved domain of the *WRKY* gene family (PF03106) was obtained from the PFAM protein family collection library (http://pfam-legacy.xfam.org/, accessed on 9 October 2022). HMMER 3.0 software was used to perform an initial search of WRKY TFs with the WRKY conserved domain within the transcriptome protein sequences of *Arabis paniculata.* To further refine the identification process, the SMART (https://smart.embl.de/, accessed on 9 October 2022) and NCBI CDD (https://www.ncbi.nlm.nih.gov/Structure/bwrpsb/bwrpsb.cgi, accessed on 9 October 2022) databases were used to verify the WRKY protein domain of *Arabis paniculate*, All datebase accessed on 11 October 2022. ApWRKY proteins lacking the conserved WRKYGQK domain were discarded, and any repetitive, redundant, or incompletely annotated sequences were removed. The remaining TFs were then confirmed as members of the WRKYs in *Arabis paniculata*.

### 4.4. Characterization Analysis of WRKY Protein Sequences in Arabis paniculata 

To further investigate the molecular structures of WRKY TFs, we employed various online tools for comprehensive characterization. The online software ExPASy-ProtParam (https://web.expasy.org/protparam/, accessed on 9 October 2022) was used to analyze the physical and chemical properties of the full-length protein, including the amino acid sequence, CDS length, molecular weight (MW), and isoelectric point (*p*I). Furthermore, the online tool SOPMA (https://npsa-prabi.ibcp.fr/cgi-bin/npsa_automat.pl?page=npsa_sopma.html, accessed on 9 October 2022) was performed to predict the secondary structure of the proteins. Additionally, subcellular localization prediction was conducted using Plant-mPLoc (http://www.csbio.sjtu.edu.cn/bioinf/plant-multi/, accessed on 9 October 2022). All datebase accessed on 5 November 2022. 

### 4.5. Phylogenetic Analysis of WRKY Genes in Arabis paniculata 

To elucidating the classification of WRKYs in *Arabis paniculata*, we conducted a multiple sequence alignment of the full-length protein sequences of these *WRKY* genes using Gene Doc software. All parameters were set to default values. Using MEGA7.0 software, we constructed a phylogenetic tree using the maximum likelihood (ML) method and a bootstrap value of 1000.

### 4.6. Motif Identification of WRKYs in Arabis paniculata 

To identify the conserved elements within the ApWRKY proteins in *Arabis paniculata*, we utilized the online software MEME (https://meme-suite.org/meme/tools/meme), accessed on 11 March 2023. A total of 10 motifs were specified, while the remaining parameters were set to default values. Subsequently, TBtools was employed to visualize the conserved motifs among the members of the ApWRKYs [49].

### 4.7. Cd Treatment

The plants were cultivated as described in Section 4.1. These plants were subjected to a Cd stress condition by watering the Hogland solution containing 5 mM CdCl_2_. Samples of the leaves were collected at 0, 0.5, 1, 6, and 24 h following Cd stress treatment. These samples were promptly frozen and stored at −80 °C in an ultra-low temperature refrigerator for further experimentation.

### 4.8. qRT-PCR Analysis of WRKY Genes in Arabis paniculata under Cd Stress

Based on the coding sequences of *ApWRKYs,* qRT-PCR primers (Appendix A) were designed using online software from the National Center for Biotechnology Information (NCBI) at https://www.ncbi.nlm.nih.gov/, accessed on 2 May 2023. Total RNA was extracted from the leaves of the sample using a kit (Aidlab, Beijing, China), and its quality was assessed using 1% agarose gel electrophoresis. The first strand of the RNA was reverse transcribed into cDNA (Cwbio, Taizhou, China). qRT-PCR assay was performed using the SuperReal fluorescence quantitative premix reagent (Tiangen, Beijing, China). *Actin* was used as an internal reference [50], and the relative expressions of the target genes were calculated using the 2^−∆∆CT^ formula [51].

### 4.9. Statistical Analysis

The data were analyzed using SPSS Statistics software (version 21, IBM, Chicago, IL, USA). After confirming the normality and homogeneity of the data, variance analysis was performed on each group. All experimental results were expressed as standard deviation (SD) (*n* = 3). A minimal significant difference test (*p* < 0.05) was conducted a priori to compare differences between different methods.

## 5. Conclusions

In this study, we obtained the full-length transcriptome of *Arabis paniculata* using three-generation sequencing technology. Subsequently, we identified the WRKY family and examined their response to Cd stress. Our findings revealed that *ApWRKY23* and *ApWRKY34* exhibited a high level of responsiveness to Cd, suggesting their potential as valuable genetic resources for developing Cd-resistant plants. Overall, this work establishes a crucial groundwork for investigating the genetic information of *Arabis paniculata* and offers significant insights for the exploration of Cd-responsive genes.

## Figures and Tables

**Figure 1 plants-12-03779-f001:**
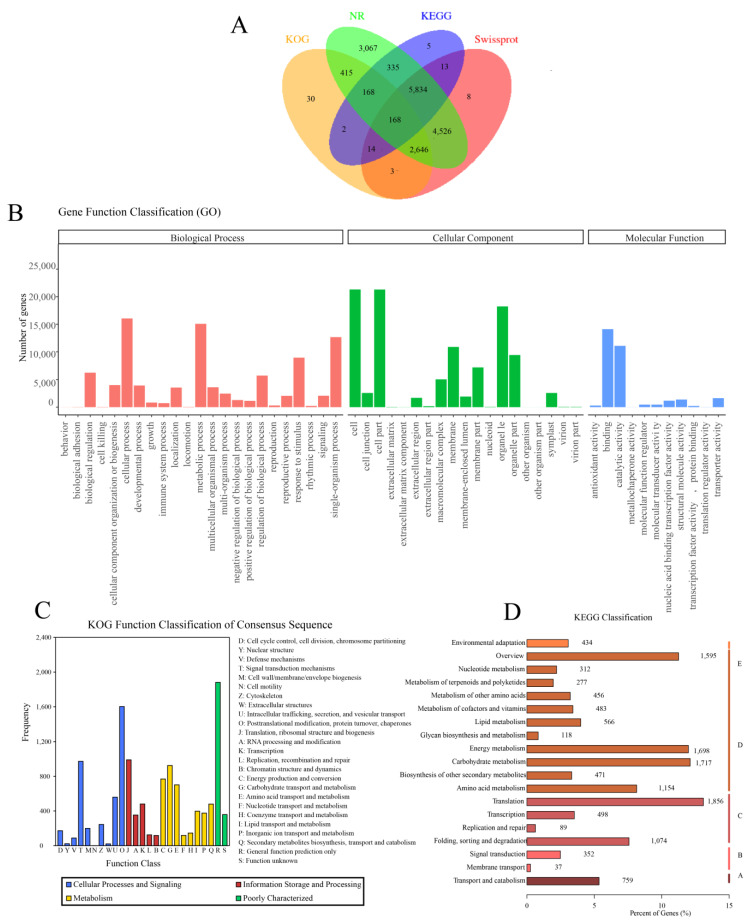
The Venn diagram of the isoform numbers annotated to four databases (**A**). GO annotation secondary classification statistics of *Arabis paniculata* (**B**). KOG classification statistics (**C**). KEGG annotation classification diagram (**D**). The data constructed by the figure (**A**–**D**) are based on the three-generation full-length transcriptome data.

**Figure 2 plants-12-03779-f002:**
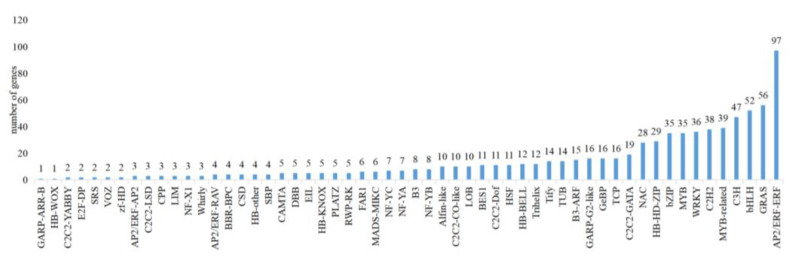
Distribution of TFs families predicted by all isoforms.

**Figure 3 plants-12-03779-f003:**
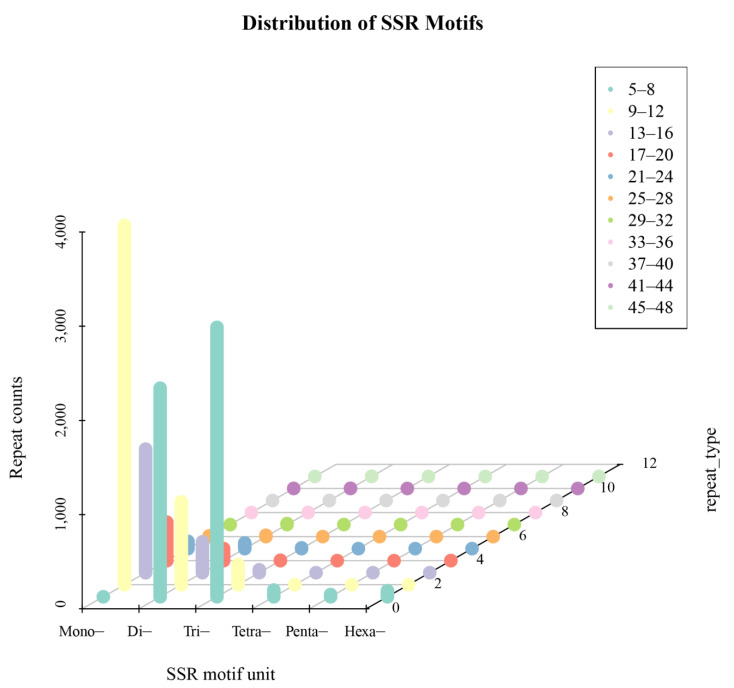
SSR tandem repeat unit type proportion statistics.

**Figure 4 plants-12-03779-f004:**
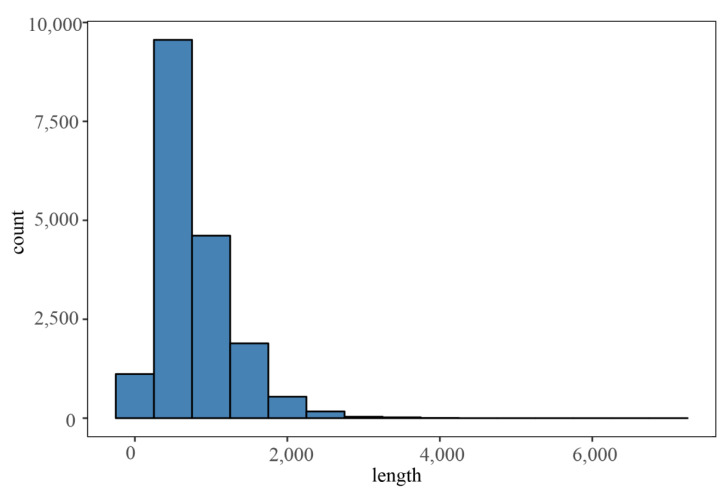
LncRNA length distribution map.

**Figure 5 plants-12-03779-f005:**
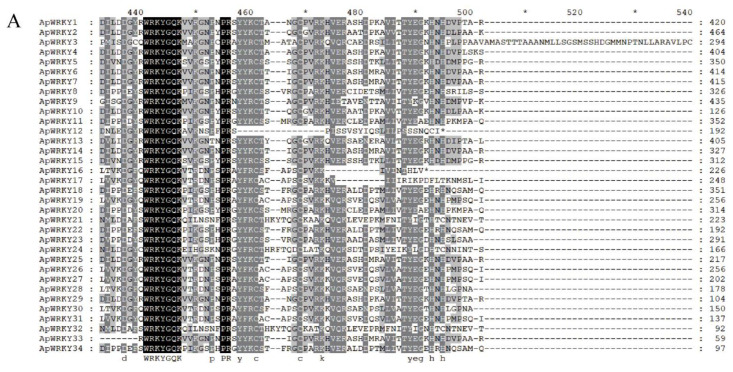
Multiple sequence alignment of WRKY protein domain in *Arabis paniculate*. * indicates a gap of 20 nucleotides. (**A**). The phylogenetic relationship triangle of *WRKY* gene family proteins in *Arabis paniculata* and *Arabidopsis thaliana*. The triangle and the star represent *Arabidopsis thaliana* and *Arabis paniculata*’s WRKY family proteins, respectively (**B**).

**Figure 6 plants-12-03779-f006:**
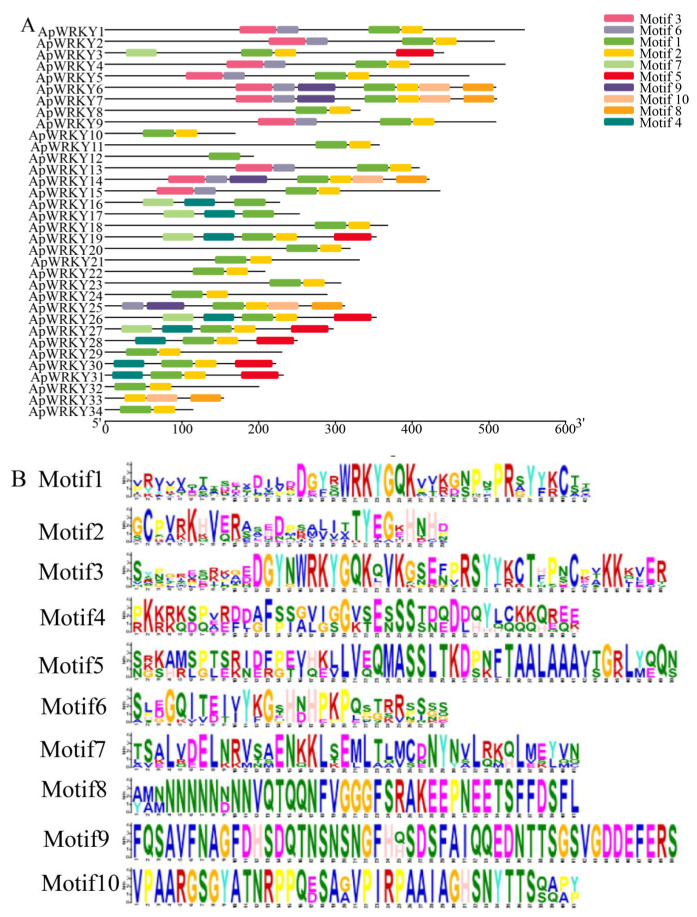
Identification of motifs of WRKY family members in *Arabis paniculata* (**A**). Distribution of conserved motifs in WRKY family of *Arabis paniculata* (**B**).

**Figure 7 plants-12-03779-f007:**
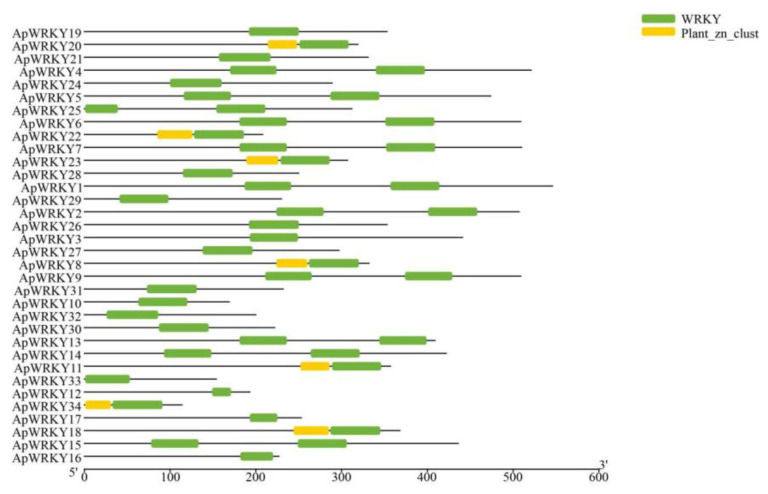
Conserved domains of the ApWRKY family.

**Figure 8 plants-12-03779-f008:**
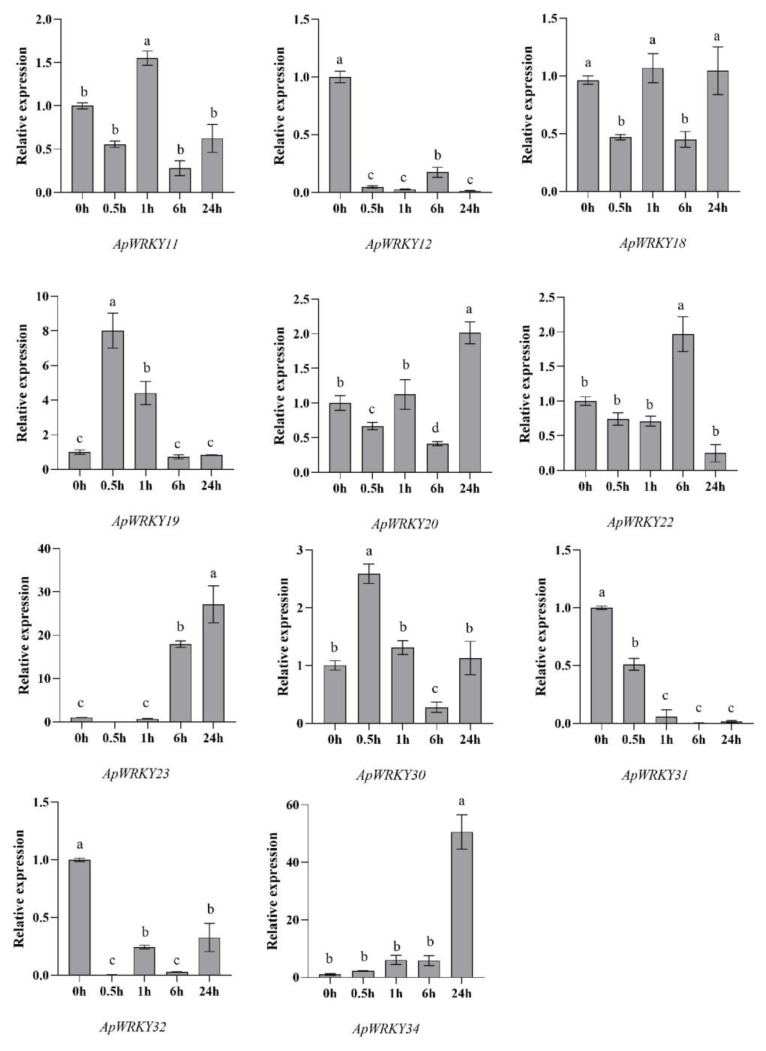
Expression levels of *ApWRKYs* at different times under 5 mM Cd stress. Bars represented by the same letters are not significantly different at *p* < 0.05.

## Data Availability

The datasets utilized in this work are available upon reasonable request.

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
