# Peer review of "Full-Length Transcriptome Sequencing Analysis and Characterization of WRKY Transcription Factors Responsive to Cadmium Stress in Arabis paniculata"

_plants, 2023, doi:10.3390/plants12213779_

Round 1
Reviewer 1 Report
Comments and Suggestions for Authors
The manuscript has been correct, seems very good.
L. 37 - Cd is not nutrient for plants and human ….
You can mention about stress plant under phytoremediation
for instances:
Bączek-Kwinta R., Antonkiewicz J., Łopata-Stasiak A., Kępka W. 2019. Smoke compounds aggravate stress inflicted on Brassica seedlings by unfavourable soil conditions. Photosynthetica, 57, 1, 1-8. DOI: 10.32615/ps.2019.026
Author Response
Comment: L. 37 - Cd is not nutrient for plants and human ….
You can mention about stress plant under phytoremediation, for instances: Bączek-Kwinta R., Antonkiewicz J., Łopata-Stasiak A., Kępka W. 2019. Smoke compounds aggravate stress inflicted on Brassica seedlings by unfavourable soil conditions. Photosynthetica, 57, 1, 1-8. DOI: 10.32615/ps.2019.026
Response: According to this comment, we revised the description in the section of introduction (Line 37-43, marked in red). At the same time, we followed the reviewer’s comment and cited the important literature recommended by the reviewer (reference number: 4). Thanks.

Reviewer 2 Report
Comments and Suggestions for Authors
Review of the manuscript plants-2684668
Title: “Full-length transcriptome sequencing analysis and characterization of WRKY transcription factors responsive to cadmium stress in Arabis paniculata”
The authors describe the molecular characterization of the family of transcription factors WRKY in the cadmium, zinc, and Lead Hyperaccumulator plant Arabis paniculate.
The work is very similar to that by Zhang et al. ( Zhang R, Chen Z, Zhang L, Yao W, Xu Z, Liao B, Mi Y, Gao H, Jiang C, Duan L and Ji A (2021) Genomic Characterization of WRKY Transcription Factors Related to Andrographolide Biosynthesis in Andrographis paniculata. Front. Genet. 11:601689. doi: 10.3389/fgene.2020.601689) Only the stress varies, here we have treatments with an excess of Cd, in a paper by Zhang we have induction of secondary metabolites.
Some information is missing. For example, how many chromosomes has the plant? Is it possible to establish a synteny map with plants other than Arabidopsis regarding the family of transcription factors WKRY? Are there many splicing variants in the different genes the Authors found to be transcription factors WKRY?
One reference is in Chinese, n°8
Liu, Z.; Zhou, L.; Gan, C.; Hu, L.; Pang, B.; Zuo, D.; Wang, G.; Wang, H.; Liu, Y., Transcriptomic analysis reveals key genes and 494 pathways corresponding to Cd and Pb in the hyperaccumulator Arabis paniculata. Ecotoxicology and Environmental Safety 2023, 495 254, 114757.
Please find another one in English.
Unfortunately, the Conclusions are a copy of the Abstract.
Apart from these observations, the work is clear and interesting.
Comments on the Quality of English Language
The English language is mostly clear and concise.
Author Response
Comment: The work is very similar to that by Zhang et al. ( Zhang R, Chen Z, Zhang L, Yao W, Xu Z, Liao B, Mi Y, Gao H, Jiang C, Duan L and Ji A (2021) Genomic Characterization of WRKY Transcription Factors Related to Andrographolide Biosynthesis in Andrographis paniculata. Front. Genet. 11:601689. doi: 10.3389/fgene.2020.601689) Only the stress varies, here we have treatments with an excess of Cd, in a paper by Zhang we have induction of secondary metabolites.
Response: Thank you very much for your professional recommendations. According to this comment, we compared the our work and the paper by Zhang et al. (2021). We have added to the discussion section of the manuscript to provide a more comprehensive understanding of gene families (Line 353-357, marked in red). Thanks.
Comment:Some information is missing. For example, how many chromosomes has the plant? Is it possible to establish a synteny map with plants other than Arabidopsis regarding the family of transcription factors WKRY? Are there many splicing variants in the different genes the Authors found to be transcription factors WKRY?
Response: Thank you for your valuable and professional feedback. As you mentioned, the full-length transcriptome sequencing of Arabis paniculata was hindered by the lack of a reference genome, resulting in the unavailability of specific location information and accurate annotation data. This limitation affected the presentation of certain graphics and hindered gene location and collinearity analysis on chromosomes. Furthermore, the absence of splicing variant and transcription factor analysis prevented the determination of multiple splicing variants in different genes of the WKRY transcription factor . Thank you.
Comment: One reference is in Chinese.
Response: According to this comment, we have replaced the Chinese reference cited in the paper by an English paper (No. 9 of the references). Thanks.
Comment:the Conclusions are a copy of the Abstract
Response: We would like to express our sincere gratitude for the valuable and professional advice and feedback you have provided. According to this comment, we have re-written the section of conclusion to make the manuscript more logical and complete (Section of Conclusion). Thanks.

Round 2
Reviewer 2 Report
Comments and Suggestions for Authors
The manuscript has been greatly improved by the corrections of the Authors. They have answered all my queries.
Author Response
Thank you !